# Determinants of Podoconiosis in Bensa District, Sidama Region, Ethiopia: A case control study

Melaku Hailu[1], Nana Chea[2], Musa Mohammed Ali[3]*, Mesay Hailu[4]

1 Tulla Primary Hospital, Hawassa, Ethiopia, 2 School of Public Health, College of Medicine and Health Sciences, Hawassa University, Hawassa, Ethiopia, 3 School of Medical Laboratory Science, College of Medicine and Health Sciences, Hawassa University, Hawassa, Ethiopia, 4 Ethiopian Public Health institute, Addis Ababa, Ethiopia

* ysnmss@yahoo.com

## Abstract

### Background

Podoconiosis is one of the neglected tropical diseases (NTD) with the greatest potential for elimination. Despite its public health importance, podoconiosis is a poorly understood disease which led to a widespread misconception about its cause, prevention, and treatment. Even though the exact global burden is still to be measured, it is estimated that at least 4 million people are affected with podoconiosis worldwide, of which more than 1.5 million people are in Ethiopia. The objective of this study was to identify the determinants of podoconiosis in Bensa District, Sidama Regional State, Ethiopia.

### Methodology/Principal findings

A community-based unmatched case-control study was used to identify the determinants of podoconiosis. The sample size was estimated using the double population proportion formula. An interviewer-administered structured questionnaire was used for data collection. Blood specimens collected from cases were tested by Filariasis Test Strip to exclude lymphatic filariasis. Data were checked for completeness, coded and entered into Epi-data Version 4.6, and exported to the SPSS version 22 software. Variables with a $p<0.2$ in the bivariate analysis were further analyzed using multivariable binary logistic regression. Multivariable logistic regression analysis was used to examine determinants that could be associated with podoconiosis with a 95% confidence interval. A total of 459 (153 cases and 306 controls) participants were included with a response rate of 100%. Factors such as the age of participant [AOR = 0.34, 95% CI (0.13–0.87)], being female [AOR = 2.90, 95% CI (1.40–6.10)], age at which shoe wearing started [AOR = 0.7, 95% CI (0.03–0.16)], not wearing shoe daily [AOR = 2.26, 95% CI (1.05–4.86)], wearing hard plastic shoe [AOR = 3.38, 95% CI (1.31–10.89)], and family history with a similar disease (leg swelling) [AOR = 10.2, 95% CI (3.97–26.37)] were significantly associated with the occurrence of podoconiosis.

**Data Availability Statement:** The authors confirm that all data underlying the findings are fully available without restriction. All relevant data are within the paper and its Supporting Information files.

**Funding:** The author(s) received no specific funding for this work.

**Competing interests:** The authors have declared that no competing interests exist.

## Conclusions/Significance

The age of the participants, gender, the age at which shoe wearing started, type of shoe the participants' wear, frequency of shoe wearing, traveling barefoot, and family history with similar disease (leg swelling) were significantly associated with the occurrence of podoconiosis. Sidama regional health bureau along with non-governmental organizations working on the neglected tropical disease should plan modalities on awareness creation and comprehensive health education on shoe wearing and foot hygiene.

## Author summary

In spite of international and national effort to reduce or eliminate podoconiosis, this study revealed high number of podoconiosis cases in Bensa district, Sidama Regional State, Ethiopia. This study demonstrated shoe as one of the most important factor associated with the occurrence of podoconiosis. Majority of podoconiosis cases identified in this study were females indicating the link between being female and the disease in the study area. Some of the possible reason could be not wearing shoe or lack of awareness about the risk factor of podoconiosis. Most participants with podoconiosis had close family member with similar disease, leg swelling. In order to reduce the occurrence of podoconiosis in the study area, concerned body should focus on increasing awareness about the risk factors of the disease particularly on the importance of proper and regular shoe wearing habit. Moreover, further study is needed to decipher why significant proportion of females and participants with history of family member with similar disease were affected by podoconiosis.

## Introduction

Podoconiosis is non-infectious form of lymphoedema that occurs in tropical regions among people who walk barefoot on clay or dusty volcanic soil. Podoconiosis causes an asymmetrical swelling of the feet and lower limbs due to lymphoedema. It has a curable pre-lymphoedema phase; however, once lymphoedema is established, podoconiosis persists and may cause life-long disability [1, 2].

Podoconiosis is widely distributed in Africa, South America, and Asia. The disease has been reported in more than 20 countries, of which ten had a high burden [3, 4]. In Africa, at least ten countries have highland areas where podoconiosis is endemic. In countries where podoconiosis is endemic, it is more prevalent than Acquired Immunodeficiency disease, tuberculosis, malaria, and lymphoedema caused by Filarial worms [3, 5]. According to a 2015 report, there are more than 1.5 million podoconiosis cases in Ethiopia [5]. Podoconiosis is common in the central highlands of Ethiopia: Amhara, Oromia, and Southern Nations Nationalities and Peoples (SNNP) regional states [3–5].

Until now, there is no clear explanation for the pathogenesis and risk factors of podoconiosis. According to the existing evidence, some factors are significantly associated with the occurrence of podoconiosis; these factors include the mineral particles found in volcanic soil and genetic susceptibility to the disease [6]. Among people who often walk barefoot in podoconiosis-endemic regions, colloid-sized particles containing aluminum, silicon, magnesium, and iron may penetrate the skin and responsible for podoconiosis [6]. These particles have been

detected in macrophages residing in the lymph nodes of the lower legs and induce an inflammatory response in the lymphatic vessels, leading to fibrosis and obstruction of the vessel lumen. This leads initially to edema of the foot and the lower leg, which gradually progresses to lymphoedema; gross lymphedema with mossy and nodular changes of the skin [6].

The clinical picture and course of podoconiosis vary based on the time of presentation. The early symptom includes recurrent episodes of burning and itching of the forefoot, especially after a period of intense physical activity. Early changes that may be observed are splaying of the forefoot, increase skin marks, lower leg edema that disappears after overnight rest, hyperkeratosis with the formation of moss-like papillomata, and rigid toes. The late-stage disease is characterized by a fusion of inter-digital space and ankylosis of the inter-pharyngeal and ankle joints [1, 7].

Podoconiosis can be prevented with the application of simple and low-priced measures. In areas where podoconiosis is endemic, the majority of the community holds significant misconceptions about the cause, care, treatment, and prevention of the disease [8]. Prevention may involve the provision of health education on how to avoid or minimize exposure to irritant soils by wearing proper shoe regularly and by covering the floor of traditional huts in endemic areas [2, 8]. In the current study, we aimed to identify the determinant of podoconiosis among people residing in Bensa district of Sidama Regional State, Ethiopia.

## Methods

### Ethics statement

Ethical clearance was obtained from Hawassa University College of Medicine and health sciences Institutional Review Board (Reference number: IRB/009/13). Before recruiting the study participants, written informed consent was obtained from participants. Formal consent was obtained from the parent or guardian for child participants. Participants with podoconiosis were linked to Hawassa University Comprehensive Specialized Hospital for better management.

### Study area

This study was conducted in Bensa district located in Sidama Regional State. The district's average altitude is 1,914 meters above sea level with an average rainfall of 1,251.2 mm annually. According to the Bensa district Health Office report of 2019, the total population of the district is 189,047 of which 92,444 were males and 96,603 were females. The district has a total of 24 kebeles, of which 22 are rural and 2 of them are urban kebeles. The district has one primary hospital and seven health centers. The majorities of the population are farmers and belong to Sidama Ethnicity. The study site was selected because of the high burden of podoconiosis cases, absence of study related to podoconiosis, and its high altitude as compared to other district of Sidama Regional State.

### Study design and period

A community-based unmatched case-control study was conducted in Bensa district from January 1 to May 28, 2021.

### Cases

Individuals with clinically confirmed podoconiosis by trained Nurses and those who were negative for filariasis.

### Controls

Individuals who resided in the selected kebeles during the study period and did not demonstrate any clinical signs and symptoms of podoconiosis.

### Eligibility criteria

All members in the selected district whose age greater than or equal to > 15 years and resided in the district during the data collection period were included. Individuals who resided in the district for less than 10 years were excluded from the study.

### Study population

All individuals living in the district whose age was greater than or equal to 15 years were included. The study population was all individuals whose age was greater than or equal to 15 years and resided in the selected kebeles (the lowest level of government administrative structure in Ethiopia). The required sample size was determined by considering factors associated with the occurrence of podoconiosis using double population proportion formula [9, 10]. The sample size was determined using Epi-info software based on the following assumptions: 95% confidence interval, 80% power, control-to-case ratio of 2:1, 9.7% participant without shoe among cases (P1), and 2.5% participant without shoe among control (P2). After considering 10% for the non-response rate, the final sample size was 459. The sample size was categorized into two: 306 for controls and 153 for cases.

### Sampling procedure

Two-stage sampling technique was employed to select the respondents. Out of the total 22 rural kebeles in the district, seven kebeles were selected using simple random sampling technique. The sample size was allocated proportionally to each selected kebeles based on the number of households in the kebeles. Health extension workers (HEWs) conducted house-to-house visits in the selected kebeles to identify persons with lymphedema. 489 clinically confirmed podoconiosis cases by nurses were used as a sampling frame for the selection of study participants. Podoconiosis cases included in this study were selected by a systematic random sampling method. Controls were also selected by systematic random sampling method from residents in the same study setting. The list of all households excluding those with podoconiosis was used as a sampling frame for selecting the control. Interval ($k^{th}$) was determined by dividing the number of households by the sample size allocated for each kebele. Every $k^{th}$ household was selected until the required sample size was obtained from each selected kebele. Whenever the selected household was not available next household was replaced (Fig 1).

### Data collection

Data were collected using an interviewer-administered structured questionnaire. The questionnaire includes socio-demographic factors, behavioral factors, genetic factors, and housing conditions. A questionnaire was prepared after reviewing similar studies conducted on podoconiosis [11–13]. Before use, the questionnaire was modified to fit the purpose of the current study. The questionnaire was first prepared in English then translated to the local language and pre-tested to ensure consistency. Six nurses for data collection, two laboratory technologists for blood testing, and one health officer for supervision were recruited and trained. Data were collected using local language (Sisaam-Afoo). For participant with clinical signs and symptoms of podoconiosis, a blood specimen was collected and tested for Filiral antigen to rule out lymphatic filariasis. For the blood test, Alere Filarial Test Strip (FTS) within inbuilt

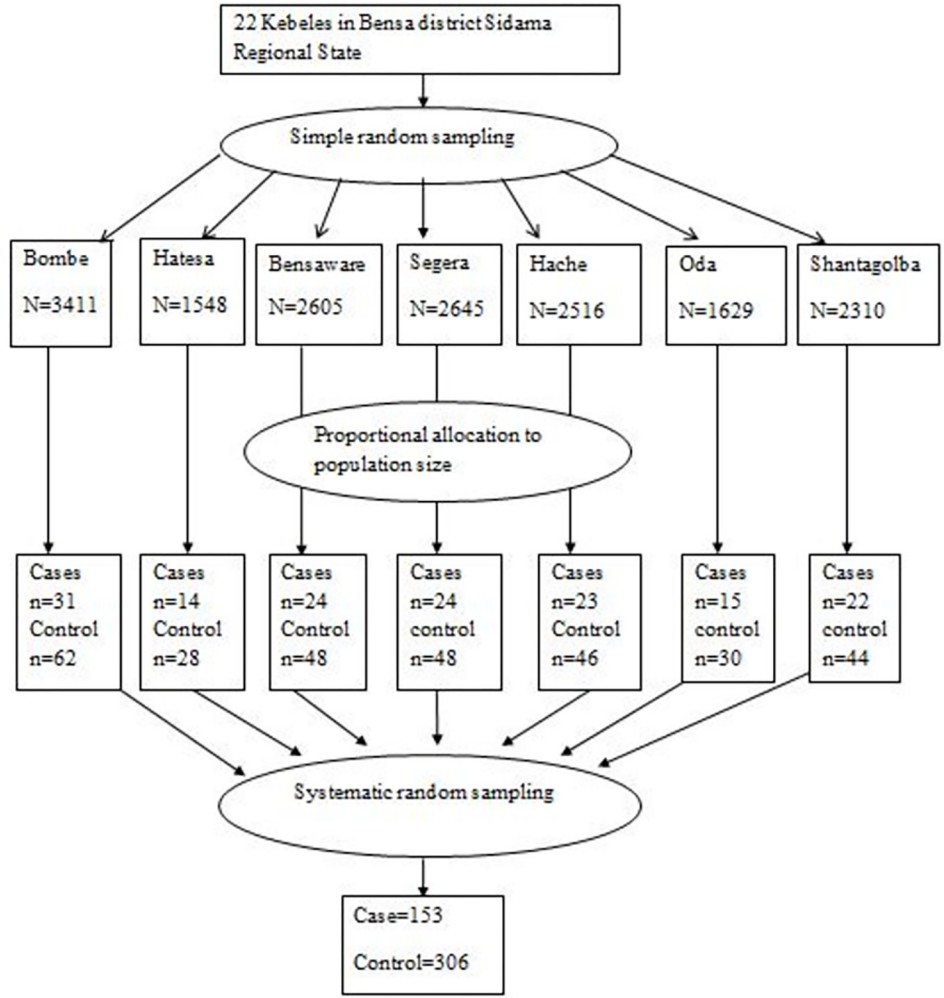

**Fig 1. Proportional distributions of sample size to each kebele.**

positive and negative control was used following manufacturer instruction [14]. The geographic locations of the participants were collected using the Geographical positioning system (GPS).

## Blood collection and Laboratory diagnosis

The blood test was performed to exclude lymphatic filariasis using FTS (Alere, Scarborough, ME). The participant's ring finger on the left hand for a right-handed person or the ring figure on the right hand for a left-handed person was cleaned with 70% alcohol and punctured using a sterile lancet. 75-μl of blood specimen was collected into the capillary tube and transferred to the pad on the FTS card. The test result of the FTS card was read at 10 minutes exactly not before, and not after. The test was interpreted as a positive if two lines appeared and as a negative if only one single line appeared.

## Data quality control

To maintain the quality of data, the questionnaire was prepared in English and then translated into the local language (Sidaam-Afoo) and then back to English to check the consistency of the

tool. A pretest was conducted among a population representing 5% of the sample size before the actual study. Feedback was used to improve the questionnaire. Alere Filarial Test Strip (FTS) was checked using known positive and negative blood specimens before it was applied for the actual study. Known positive and Negative blood specimens were used to check the performance of the test strip. Before use, the expiry date of the test strip was checked. Moreover, the manufacturer manual and inbuilt quality control were followed during a blood test. Data were checked for completeness and consistency by the supervisor. The supervisor made a close follow-up and assistance during data collection. Prior to the data collection, training was given for data collectors and the supervisor about the objectives of the study, diagnosis of podoconiosis, data collection instruments, and ethical considerations. Moreover, clinical diagnosis of podoconiosis was made by nurses certified by national podoconiosis action network.

## Data processing and analysis

The data were checked for completeness and consistency. The data were entered using Epidata version 4.6 and exported to SPSS-version 22. The results were presented using descriptive statistics and ratios. Variables with a $p<0.2$ in the bivariate analysis were further analyzed by multivariable binary logistic regression. Multivariable binary logistic regression analysis was used to examine the determinants of podoconiosis. Odds ratios and 95% confidence interval were used to measure the strength of the association. The geographic map that shows the distribution of the disease was drawn using ArcGIS 10.4.

## Operational definitions

**Podoconiosis case.**    An individual with bilateral but asymmetric swelling which develops first in the foot often confined to the lower leg and showed negative results for Filariasis Test Strip (FTS).

**Control.**    An individual living in the selected kebeles and clinically did not demonstrate clinical signs and symptoms of podoconiosis.

**Proper foot hygiene.**    If patients wash their feet daily with soap and apply lotion after proper cleaning of the foot it will be considered as 'proper foot hygiene'.

**Family history of podoconiosis.**    If the patient has one or more close relatives with podoconiosis.

## Results

### Socio-demographic characteristics

A total of 459 participants were included in this study with a response rate of 100%. Of the total participants, 153 were cases and 306 were controls. Ninety-three (60.8%) of cases were females whereas 201 (65.7%) of controls were males. The mean ages (±SD) of cases and controls were 42.85(±17.426) and 38.82(±13.435) years respectively. Ninety-eight (64.1%) of cases and 120 (39.2%) of controls had no formal education and eighty (52.3%) of cases and 188 (61.4%) of controls were farmers. 123 (80.4%) of cases and 281 (91.8%) of controls were married. The median monthly income was 400 (IQR, 400–600) and 500 (IQR, 300–700) Ethiopian birr for cases and controls respectively (**S1 Table**).

### Shoe-wearing and foot care practice

Of the study participants, 128 (83.7%) cases and 300 (98%) controls wear shoe at least once in their lifetime. The mean age (±SD) at which shoe-wearing started was 16.66 (± 9.18) and 9.85 (±5.27) years for cases and controls respectively. During the interview, 100 (78.1%) of cases

and 278 (92.7%) of controls wore shoe. Seventy-one (55.5%) of cases responded that they did not wear shoe daily, whereas 219 (70.7%) of control reported that they wear shoe daily. Ninety-five (74.2%) of cases and 285 (95%) of controls did not travel barefoot for different social purposes. 85 (55.6%) of cases and 173 (56.9%) of controls wash their feet once per day. The majority 116 (75.8%) of cases and 173 (56.5%) controls washed their feet by using water only (**S2 Table**)

### Leg swelling history, FTS card result, and physical examination

All 153 (100%) cases had negative results for the FTS test and swelling started below the knee. There was no size difference between the two legs in 99 (64.9%) of cases. More than half (n = 89; 58.2%) cases had soft and water bag (pitting) type of swelling whereas 64 (41.8%) had the hard and fibrotic type of leg swelling. Eighty-seven (56.9%) of cases experienced acute attacks in the last six months. In this study, five clinical stages of podoconiosis were identified with stage II (n = 51; 33.3%) being the most common followed by stage III (n = 40; 26.1%), stage I (n = 34; 22.2%), stage IV (n = 13; 8.5%), and stage V (n = 15; 9.8%).

### Geographical distribution of cases

A total of 489 people with podoconiosis were identified by house-to-house visits in seven selected kebeles. The highest numbers of cases were identified in Bombe, Bensaware, and Shantagolba kebeles. After the identification of cases, we collected the geographic locations of the patients using the Geographical positioning system (GPS) and generated a spatial distribution of cases at the village label (**Fig 2**).

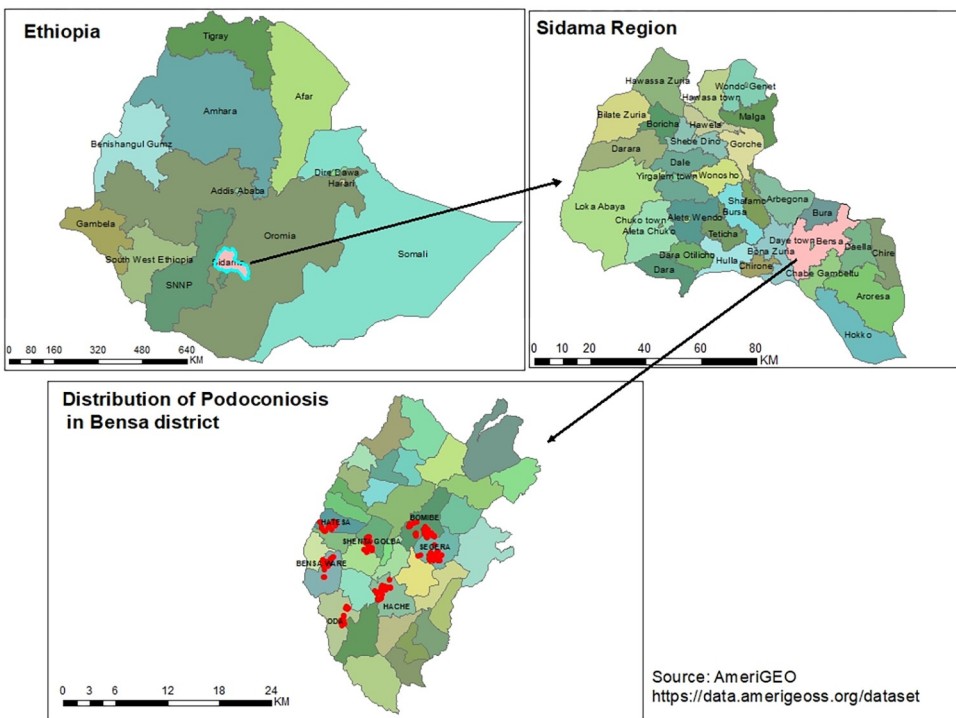

**Fig 2. Spatial patterns of village-label distribution of podoconiosis cases in Bensa district, Sidama Regional state, Ethiopia 2021.** The country or region border shape file is obtained from website: https://data.amerigeoss.org/dataset.

## Bivariate analysis

In this study; age ($p$ = 0.002), gender ($p<0.0001$), educational status ($p<0.0001$), occupation ($p$ = 0.001), marital status ($p$ = 0.003), floor type (p = 0.009), age at which shoe wearing started ($p<0.0001$), shoe wearing status during the time of interview ($p<0.0001$), type of shoe the participants use ($p<0.0001$), frequency of shoe wearing ($p<0.0001$), walking barefoot for different social purposes ($p<0.0001$), duration of foot wash ($p$ = 0.094), foot washing practice ($p<0.0001$), frequency of shoe wearing ($p<0.0001$), shoe wearing during farming ($p<0.0001$), and having a family with a history of similar diseases ($p<0.0001$) had a $p<0.2$ in bivariate binary logistic analysis and they were candidate for multivariable analysis (**Table 1**).

## Multivariable logistic analysis

Independent variable that had a $p<0.2$ during bivariate analysis were further analyzed using multivariable logistic regression. After multivariable binary logistic analysis, age, gender, frequency of shoe wearing, age at which shoe wearing started, type of shoe the respondent use, barefoot walking for different social purposes, and history of a family with leg swelling were significantly associated with podoconiosis.

The odds of podoconiosis in individuals within 35–44 years age group were lower by a factor of 0.34 compared to individuals >55 years old [AOR = 0.34, 95% CI (0.13–0.87)]. The odds of developing podoconiosis among female participants were higher by a factor of 2.9 than among male participants [AOR = 2.90, 95% CI (1.40–6.10)]. The odds of developing podoconiosis among those who started wearing shoe at age of 2–10 years were lower by a factor of 0.07 than those who started shoe-wearing after 10 years [AOR = 0.07, 95% CI (0.03–0.16)]. The odds of developing podoconiosis among those who didn't wear shoe daily were higher by factor 2.26 than those who wear shoe daily [AOR = 2.26, 95% CI (1.05–4.86)]. The odds of developing podoconiosis among those who wear hard plastic shoe were higher by factor 3.38 than those who wear canvas shoe [AOR = 3.38, 95% CI (1.31–10.89)]. The odds of podoconiosis in participants who regularly walk barefoot for different social purposes were higher by factor 3.38 as compared to those who walk barefoot for different social purposes [AOR = 11.6,95% CI (3.86–35.12)]. The odds of developing podoconiosis in participants who have a history of a family with leg swelling were higher by factor 10.2 than those who have no history of a family with leg swelling [AOR = 10.2, 95% CI (3.97–26.37)] (Table 2).

## Discussion

In this study, we have included 153 cases and 306 controls. Factors such as age, gender, age at which shoe wearing started, type of shoe, frequency of shoe wearing, walking barefoot for different social purposes and history of a family with a similar disease were significantly associated with podoconiosis.

The odds of podoconiosis among participants aged above 55 years were 66% higher than those within the age group of 35–44 years. This finding is consistent with the epidemiological study conducted in Ethiopia [15, 16]. This might be due to the shoe-wearing habits of older individuals in the district. The high prevalence of podoconiosis among the old age group could also be explained by the prolonged duration between the time of exposure and disease development that leads to the accumulation of cases at an older age.

In our study, the odds of podoconiosis among females were higher by a factor of 2.9 than male participants. Similar finding was reported by studies conducted in Guliso and Bedele, Ethiopia [15, 17, 18]. This might be because women are less likely to have shoes in many rural parts of Ethiopia. However, the finding of the current study is not in line with the study conducted in Northern parts of Ethiopia where females are less likely to be affected by

**Table 1. Bivariate analysis of determinants of podoconiosis Bensa district, Sidama Regional state, Ethiopia 2021.**

| Characteristics | Category | Cases n (%) | Control n (%) | COR (95% CI) | p-value |
|---|---|---|---|---|---|
| Gender | Male | 60(39.2) | 201(65.7) | 1 | |
| | Female | 93(60.8) | 105(34.3) | 2.97(1.99–4.43) | <0.0001** |
| Age | 15–24 | 18(11.8) | 23(7.5) | 0.77(0.36–1.61) | 0.481 |
| | 25–34 | 37(24.2) | 110(35.9) | 0.33(0.19–0.58) | <0.0001* |
| | 35–44 | 33(21.6) | 88(28.8) | 0.37(0.20–0.66) | 0.001* |
| | 45–55 | 21(13.7) | 42(13.7) | 0.49(0.25–0.96) | 0.037* |
| | ≥55 | 44(15) | 43(7.5) | 1 | |
| Occupation | Merchant | 5(3.3%) | 25(8.2) | 0.71(0.2–2.56) | 0.605 |
| | Farmer | 80(52.3) | 188(61.4) | 1.52(0.63–3.66) | 0.350 |
| | Housewife | 40(26.1) | 46(15) | 3.11(1.21–7.94) | 0.018* |
| | Have no jobs | 21(13.7) | 22(7.2) | 3.41(1.22–9.55) | 0.020* |
| | Other | 7(4.6) | 25(8.2) | 1 | |
| Educational status | No education | 98(64.1) | 120(39.2) | 4.9(1.40–17.12) | 0.013* |
| | Read and write | 11(7.2) | 40(13.1) | 1.65(0.41–6.64) | 0.481 |
| | Grade 1–8 | 32(20.9) | 94(30.7) | 2.04(0.56–7.39) | 0.277 |
| | Grade 9–12 | 9(5.9) | 34(11.1) | 1.59(0.38–6.61) | 0.525 |
| | Above grade 12 | 3(2) | 18(5.9) | 1 | |
| Marital status | Single | 20(13.1) | 17(5.6) | 0.39(0.09–1.69) | 0.208 |
| | Married | 123(80.3) | 281(91.8) | 1.46(0.04–0.55) | 0.004* |
| | Divorced | 1(0.7) | 5(1.6) | 0.07(0.01–0.82) | 0.035* |
| | widowed | 9(5.9) | 3(1) | 1 | |
| Floor type | Earth | 132(86.3) | 234(76.5) | 0.85(0.30–2.43) | 0.756 |
| | Bamboo | 12(7.8) | 22(7.2) | 0.82(0.23–2.86) | 0.753 |
| | Cement | 3(2) | 41(13.4) | 0.11(0.02–0.52) | 0.006* |
| | other | 6(3.9) | 9(2.9) | 1 | |
| Ever worn shoe | No | 25 (16.3) | 6 (2) | 9.78(3.91–24.38) | <0.0001* |
| | Yes | 128 (83.7) | 300 (98) | 1 | |
| Age at which shoe wearing started | 2–10 years | 38 (29.7) | 214 (71.3) | 0.17(0.11–0.28) | <0.0001* |
| | >10 years | 90 (70.3) | 86 (28.7) | 1 | |
| Shoe wearing during interview | No | 28 (21.9) | 22 (7.3) | 3.534(1.94–6.47) | <0.0001* |
| | Yes | 100 (78.1) | 278 (92.7) | 1 | |
| Type of shoe used | Hard plastic | 27 (27) | 25 (9) | 2.84(1.43–5.66) | 0.003* |
| | Open sandal | 28 (28) | 45 (16.2) | 1.64(0.87–3.08) | 0.126* |
| | Leather | 15 (15) | 129 (46.4) | 0.31(0.16–0.60) | 0.001* |
| | Canvas | 30 (30) | 79 (28.4) | 1 | |
| Shoe wearing at farm | No | 77 (60.2) | 122 (40.7) | 2.20(1.44–3.36) | <0.0001* |
| | Yes | 51 (39.8) | 178 (59.3) | 1 | |
| Walking bare foot for different social reason | No | 95 (74.2) | 285 (95) | 1 | |
| | Yes | 33 (25.8) | 15 (5) | 6.6 (3.44–12.68) | <0.0001* |
| Shoe wearing frequency | Not daily | 71 (55.5) | 88 (29.3) | 3.00(1.96–4.60) | <0.0001* |
| | Daily | 57 (44.5) | 212 (70.7) | 1 | |
| Feet washing time | If they are dirty | 56 (36.6) | 98 (32) | 2.08(1.06–4.09) | 0.034* |
| | Before sleeping | 83 (54.2) | 157 (51.3) | 1.93(1.01–3.68) | 0.048* |
| | After my job | 14 (9.2) | 51 (16.7) | 1 | |
| Washing practice | By water only | 116 (75.8) | 173 (56.5) | 2.41(1.56–3.72) | <0.0001* |
| | By water and soap | 37 (24.2) | 133 (43.5) | 1 | |

(*Continued*)

**Table 1.** (Continued)

| Characteristics | Category | Cases n (%) | Control n (%) | COR (95% CI) | p-value |
|---|---|---|---|---|---|
| History of family with leg swelling | Yes | 47(30.7) | 14(4.6) | 9.25(4.89–17.48) | <0.0001* |
| | No | 106(69.3) | 292(95.4) | 1 | |

CRO: Crude Odds Ratio, CI: Confidence Interval

* p-value <0.2

** Employed and Daily laborer

*** Ceramic tiles, Dung and Wood/plank

podoconiosis than males [9]. This might be due to cultural or economic differences between the study sites.

The other predictor of podoconiosis was the type of shoe the participants use. The odds of podoconiosis among participants who wear plastic shoes were higher by a factor of 3.38 than those who wear canvas shoes. This might be due to the fact that plastic shoes offer inadequate protection against the soil [6].

The odds of podoconiosis among participants who started wearing shoe after the age of 10 years were 93% higher than those who started wearing shoe between 2–10 years of age. This finding agrees with the study conducted in Sodo, South Ethiopia, Waghmra, North Ethiopia, and Dano, Central Ethiopia [11, 16, 19]. Delay in starting shoe wearing might have increased the frequency of contact with soil which ultimately increases the chance of acquiring podoconiosis. This indicates starting of wearing shoe at an early age may have a role in reducing the occurrence of podoconiosis.

**Table 2.** Multivariable analysis on determinants of podoconiosis in Bensa district, Sidama, Regional state, Ethiopia 2021.

| Characteristics | Category | Cases (%) | Control (%) | COR (95% CI) | AOR(95%CI) | p-value |
|---|---|---|---|---|---|---|
| Gender | Male | 60(39.2) | 201(65.7) | 1 | 1 | |
| | Female | 93(60.8) | 105(34.3) | 2.97(1.99–4.43) | 2.90(1.4–6.10) | 0.004** |
| Age | 15–24 | 18(11.8) | 23(7.5) | 0.77(0.36–1.61) | 2.1(0.42–10.51) | 0.314 |
| | 25–34 | 37(24.2) | 110(35.9) | 0.33(0.19–0.58) | 0.61(0.21–1.83) | 0.369 |
| | 35–44 | 33(21.6) | 88(28.8) | 0.37(0.20–0.66) | 0.34(0.13–0.89) | 0.029** |
| | 45–54 | 21(13.7) | 42(13.7) | 0.49(0.25–0.96) | 0.59(0.20–1.76) | 0.343 |
| | >55 | 44(28.8) | 43(14.1) | 1 | 1 | |
| Age at first shoe wearing started | 2–10 years | 38 (29.7) | 214 (71.3) | 0.17(0.11–0.28) | 0.07(0.03–0.16) | <0.0001** |
| | >10 years | 90 (70.3) | 86 (28.7) | 1 | 1 | |
| Type of shoe | Hard plastic | 27 (27) | 25 (9) | 2.84(1.43–5.66) | 3.38(1.31–10.93) | 0.028** |
| | Open sandal | 28 (28) | 45 (16.2) | 1.64(0.87–3.08) | 1.86(0.69–4.99) | 0.218 |
| | Leather | 15 (15) | 129 (46.4) | 0.31(0.16–0.60) | 0.39(0.14–1.09) | 0.074 |
| | Canvas | 30 (30%) | 79 (28.4) | 1 | 1 | |
| Walk barefoot for different social reasons | No | 95 (74.2) | 285 (95) | 1 | 1 | |
| | Yes | 33 (25.8) | 15 (5) | 6.6 (3.44–12.68) | 11.6(3.86–35.12) | <0.0001** |
| Wearing frequency | Not daily | 71 (55.5) | 88 (29.3) | 3.00(1.96–4.60) | 2.26(1.09–4.86) | 0.029** |
| | Daily | 57 (44.5) | 212 (70.7) | 1 | 1 | |
| History of family with leg swelling | Yes | 83(54.2) | 9(2.9) | 9.25(4.89–17.48) | 10.2(3.97–26.37) | <0.0001** |
| | No | 70(45.8) | 297(97.1) | 1 | 1 | |

CI: Confidence interval, COR: Crude Odd Ratio, AOR: Adjusted Odd Ration, ^*statistically significant

The odds of podoconiosis in participants who walk barefoot for different social purposes were higher by factor 11.6. This finding is in agreement with the study conducted in Waghmra, North Ethiopia, and Soddo, South Ethiopia [16, 19]. Barefoot walking could have increased the chance of acquiring podoconiosis as it predisposes to irritant soil elements [1].

This study also showed that the odds of podoconiosis among participants who don't wear shoe daily to be higher by a factor of 2.26 than those who wear shoe daily. The result of this study is consistent with studies conducted in Gojam, North West Ethiopia, Danot, Central Ethiopia, Waghmra, North Ethiopia, and Kenya [9, 11, 19, 20]. Not wearing shoe regularly might have increased the exposure between the feet and irritant volcanic red clay soil which eventually increased the chance of podoconiosis.

Moreover, the odds of podoconiosis in individuals with a family history of podoconiosis were higher by a factor of 10.2 compared to those with no family history of podoconiosis. This finding is similar to the study conducted in Gojam, Ethiopia, and West Uganda [9, 21]. This may indicate the role of genetic background in the development of podoconiosis [3]. There are studies which indicated relationship between some HLA Class II variants and occurrence of podoconiosis [22, 23].

Even though podoconiosis is destined to be eliminated, the current study indicated its existence with some risk factors significantly associated with it. In addition to physical damage, podoconiosis has a social impact making it a very important public health problem to be addressed. Even though some factors are to be studied further, this study revealed that older age, using shoe appropriately, and family history with podoconiosis are predictors of podoconiosis. Finding high odds of podoconiosis among participants with a family history of podoconiosis indicates the importance of additional studies to assess the role of genetic make in the pathogenesis of podoconiosis.

## Limitation of this study

As this study used a case-control study design, there might be a chance of recall bias. In this study most control participants were older and males which can bias the finding of this study. We acknowledge that the high number of males in the control sample is likely to have led us to overestimate the association of podoconiosis with gender This occurred because we used the kebele register as the sampling frame for controls. We ended up with more male than female participants as more males were registered than females at kebele level. In this study we have categorized age to assess which age groups are more prone to podoconiosis; however, the study we have used for comparison in the discussion used age as a continuous variable.

## Conclusions

In this study factor such as age, gender, age at which shoe wearing, type of shoe participants use, frequency of shoe wearing; walking barefoot for different social purposes, and participants with a family history of podoconiosis were significantly associated with the occurrence of podoconiosis. Podoconiosis control and prevention programs should give attention to females residing in the study area to treat and rehabilitate them. Sidama regional health bureau along with non-governmental organizations working on the neglected tropical disease should plan modalities on awareness creation and comprehensive health education on shoe wearing and foot hygiene. Additional studies that are needed to address why females and participants with a family history of podoconiosis are disproportionately affected.

## Supporting information

**S1 Table. Socio-demographic characteristic of the study participants in Bensa Woreda, Sidama Regional State, Ethiopia, 2021.**
(DOCX)

**S2 Table. Shoe wearing and foot wash practice of participants in Bensa woreda, Sidama Regional state, Ethiopia 2021.**
(DOCX)

## Acknowledgments

We would like to thank all Staff off Bensa district Health office for facilitating the study. We acknowledge Health extension workers and Laboratory technologist for data collection. We would also like to thank study participants for their willingness to take part in the study.

## Author Contributions

**Conceptualization:** Melaku Hailu, Nana Chea, Mesay Hailu.

**Data curation:** Melaku Hailu, Nana Chea.

**Formal analysis:** Melaku Hailu.

**Investigation:** Melaku Hailu, Mesay Hailu.

**Methodology:** Melaku Hailu, Nana Chea, Musa Mohammed Ali, Mesay Hailu.

**Project administration:** Melaku Hailu, Nana Chea, Mesay Hailu.

**Resources:** Melaku Hailu, Mesay Hailu.

**Software:** Melaku Hailu, Mesay Hailu.

**Supervision:** Nana Chea, Mesay Hailu.

**Visualization:** Nana Chea, Musa Mohammed Ali.

**Writing – original draft:** Melaku Hailu, Musa Mohammed Ali.

**Writing – review & editing:** Melaku Hailu, Musa Mohammed Ali.

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
