## [Decision Letter · Decision Letter 0]

20 Apr 2023

Dear Dr. Ali,

Thank you very much for submitting your manuscript "Determinants of Podoconiosis in Bensa Woreda, Sidama Region, Ethiopia: A community based survey" for consideration at PLOS Neglected Tropical Diseases. As with all papers reviewed by the journal, your manuscript was reviewed by members of the editorial board and by several independent reviewers. In light of the reviews (below this email), we would like to invite the resubmission of a significantly-revised version that takes into account the reviewers' comments. 

We cannot make any decision about publication until we have seen the revised manuscript and your response to the reviewers' comments. Your revised manuscript is also likely to be sent to reviewers for further evaluation.

Sincerely,

Gail Davey

Guest Editor

Victoria Brookes

Section Editor

Thank you for submitting the manuscript "Determinants of Podoconiosis in Bensa Woreda, Sidama Region, Ethiopia: A community based survey".

While it is not acceptable for publication in its current form, PLoS NTDs is prepared to review a version revised to reflect the reviewers' comments. Please pay particular attention to Reviewer 1's comments on:

i) Representativeness of controls (a strategy to overcome possible bias has been suggested - "It may be possible to correct for age- and sex-bias of controls by re-selecting a sub-set of controls, matched with cases on age-group and gender, and re-analysing the data.");

ii) Including age as a continuous rather than categorical variable;

iii) Interpretation of ORs (again the reviewer has made clear suggestions as to how this can be improved);

iv) Existing studies on risk factors (and on clinical presentation).

Reviewer 2's comments on the use of the term 'elephantiasis', several areas of clarification in the methods, and the suggestion that some recommendations are stated, all need to be addressed.

In Addition:

The term 'podoconiosis' only needs a capital 'P' at the beginning of a sentence - think of it like for example 'measles'.

Line 278: it is misleading to state that people above a certain age are more like 'to develop' podoconiosis since it is a chronic disease that persists. What you have measured is disease prevalence within each age group, not incidence, which reflects cases of disease accumulating over time. Please rephrase.

In addition, the whole manuscript needs careful English editing for it to be accepted for publication.

Reviewer's Responses to Questions

**Key Review Criteria Required for Acceptance?**

**Methods**

-Are the objectives of the study clearly articulated with a clear testable hypothesis stated?

-Is the study design appropriate to address the stated objectives?

-Is the population clearly described and appropriate for the hypothesis being tested?

-Is the sample size sufficient to ensure adequate power to address the hypothesis being tested?

-Were correct statistical analysis used to support conclusions?

-Are there concerns about ethical or regulatory requirements being met?

Reviewer #1: -Are the objectives of the study clearly articulated with a clear testable hypothesis stated?

Yes

-Is the study design appropriate to address the stated objectives?

Yes

-Is the population clearly described and appropriate for the hypothesis being tested?

There are a few details missing for the sampling procedure section. Since the controls are intended to represent the background/ average population, we’d expect a sex ratio of around 1:1 in controls, but actually have a male:female ratio of around 2:1, suggesting that the sampling of controls was not random. For example, if only one person was selected from control households, and no randomisation was used to select within households, would it usually be the household head- therefore more likely to be male? It’s likely this bias has inflated the estimate of the association with sex. The sampling procedure may have also affected the age-distribution of controls- although this is harder to judge- only 7.5% of controls were aged 15-24, and the AOR for podo in this age group was 2.1 vs the oldest age group (with a wide CI), which is surprising. Oversampling of household heads as controls may also account for the finding that 35-44 year olds had significantly lower odds of disease while the other age groups did not.

Apart from this, data collection procedures and analysis appear robust.

-Is the sample size sufficient to ensure adequate power to address the hypothesis being tested?

Yes

-Were correct statistical analysis used to support conclusions?

It’s not clear why age was categorised rather than being handled as a continuous variable.

-Are there concerns about ethical or regulatory requirements being met?

The authors provide evidence of ethical review and give details of consent taking. 

There are no obvious ethical issues to note, although it is not stated whether cases were linked to care (or if this was not required for a specific reason).

Reviewer #2: Are the objectives of the study clearly articulated with a clear testable hypothesis stated? to some extent

-Is the study design appropriate to address the stated objectives? yes

-Is the population clearly described and appropriate for the hypothesis being tested? yes

-Is the sample size sufficient to ensure adequate power to address the hypothesis being tested? yes

-Were correct statistical analysis used to support conclusions? yes but some has very wide CI

-Are there concerns about ethical or regulatory requirements being met? No

Background 

The word elephantiasis is derogatory for the affected peoples thus it is better replaced by lymphoedema.

Line 53-55 This sentence should be rephrased for better clarity ‘In parts of these countries where it is endemic, Podoconiosis is more than Acquired Immunodeficiency disease, tuberculosis, malaria, and elephantiasis caused by Filarial worms [3, 5]’. Moreover, if it is about prevalence, it must be re-checked for the accuracy of the data. 

Methods 

The selection of this study site should be better justified than saying ‘it is high altitude as compared to other Wordas of Sidama Regional State’. As I think most of the areas in Sidama region have high altitude.

In the design why you prefer unmatched case-control study instead of matching ??

What was the role of trained health extension worker (HEW) in this study? Is it screening and linking to nurses ? as diagnosis of cases by HEW/community health workers is unlikely 

Have you included unaffected family members of cases as controls? If you do so it is difficult to assess some determinants

Line 131 &336 -The word ‘laboratory scientists’ should be replaced by laboratory technician/technologist

Line 133- The ‘Filiral worm’ should be replaced by filarial antigen

Line181- The sub topic ‘Ethical approval’ should be before the sub-topic ‘operational definitions’

Line188- a response rate of 100% is usually unexpected, how do you achieve it? It is better if you justify the absence of inducement.

**Results**

-Does the analysis presented match the analysis plan?

-Are the results clearly and completely presented?

-Are the figures (Tables, Images) of sufficient quality for clarity?

Reviewer #1: -Does the analysis presented match the analysis plan?

Yes

-Are the results clearly and completely presented?

Page 29- odds ratios should all be interpreted in terms of odds, rather than risk (e.g. “Participants in age groups 35-44 years were 0.34 times less likely to develop Podoconiosis”). The correct interpretation for an OR of 0.34 is “The odds of podo for individuals 35-44 was 64% lower than those for individuals aged >55” or “the odds of podo in individuals 35-44 were lower by a factor of 0.34 compared to individuals >55”. Since ORs of less than one are difficult to interpret, I usually suggest leaving it as “the odds of X were lower for individuals in group A compared to group B and this difference was statistically significant”. Same comment for results lines 259 , 266 and 296, and also in the discussion lines 278, 294, 300 and 309. For all of these, it is valid to say “more likely” or “less likely” without giving the magnitude. 

-Are the figures (Tables, Images) of sufficient quality for clarity?

Yes

Reviewer #2: Does the analysis presented match the analysis plan? yes

-Are the results clearly and completely presented? yes

-Are the figures (Tables, Images) of sufficient quality for clarity? yes, with some stated comments

Results 

Line203- The sentence in this line is not necessary , the earlier sentence (Line202) is sufficient. 

In Table 2- Type of shoes-canvas among control groups should be 28.4 % ( not 284)

Line214- Was there involvement of physicians/ dermatologists in either for the training of nurses or conducting of physical examinations including podoconiosis staging ?

Line232- You have used the term bivariate analysis, where as in the abstract (Line30) you used bivariable analysis. The same is true in Line-239. Be consistent, bivariate is preferable.

Line241- same comment as Line 232

Line 248 you mentioned Multivariate logistic analysis, where as in abstract (L30) you used multivariable analysis. Be consistent, multivariable is preferable.

Lines 250-251- same as Line248 comment

Line255- the word ‘0.34 times less likely’ looks vague. You better say less likely. 

Line-259- same comment as Line255

Line260- it should be 0.07, not 07

Line271- same comment as Line248

In table 4- as it is a multivariable analysis it is not necessary to mention p <0.2. This is only relevant in the bivariate analysis.

Discussion 

Generally, there are some linguistic errors which affected the flow of information

Line280-The phrase ‘epidemiological study conducted in Ethiopia and Soddo Zuria district’ should be re-written correctly.

Lines280-281- the increase in odds of podoconiosis in the elderly might also be explained by the prolonged duration between the time of exposure and disease development ?

Lines 285-286- Do you have a reference that shows a difference in preventive behaviours between men and women?

Line302- Barefoot walking, not barefoot waling 

Line312- you can add more references showing the role of genes in occurrence of podoconiosis, including Tekola-Ayele et al ‘HLA Class II Locus and Susceptibility to Podoconiosis’. Moreover, a recent nature article on ‘Replication of HLA class II locus association with susceptibility to podoconiosis in three Ethiopian ethnic groups’

**Conclusions**

-Are the conclusions supported by the data presented?

-Are the limitations of analysis clearly described?

-Do the authors discuss how these data can be helpful to advance our understanding of the topic under study?

-Is public health relevance addressed?

Reviewer #1: -Are the conclusions supported by the data presented?

Yes

-Are the limitations of analysis clearly described?

No

-Do the authors discuss how these data can be helpful to advance our understanding of the topic under study?

No

-Is public health relevance addressed?

No

Reviewer #2: Are the conclusions supported by the data presented? yes

-Are the limitations of analysis clearly described? to some extent 

-Do the authors discuss how these data can be helpful to advance our understanding of the topic under study? no

-Is public health relevance addressed? to some extent

Conclusion 

The conclusion should be followed by recommendations. I think that is the main reason for conducting this study.

**Editorial and Data Presentation Modifications?**

Reviewer #1: The introduction gives a clear and accessible background on the pathogenesis, clinical manifestations and control of podoconiosis, providing appropriate references and explaining where understanding remains incomplete. However it does not reference any previous evidence relevant to the main question of the study- i.e. risk factors. I suggest including references such as 

* Deribe, K., Brooker, S.J., Pullan, R.L., Sime, H., Gebretsadik, A., Assefa, A., Kebede, A., Hailu, A., Rebollo, M.P., Shafi, O. and Bockarie, M.J., 2015. Epidemiology and individual, household and geographical risk factors of podoconiosis in Ethiopia: results from the first nationwide mapping. The American journal of tropical medicine and hygiene, 92(1), p.148.

and

Molla, Y. B., Le Blond, J. S., Wardrop, N., Baxter, P., Atkinson, P. M., Newport, M. J., & Davey, G. (2013). Individual correlates of podoconiosis in areas of varying endemicity: a case-control study. PLoS neglected tropical diseases, 7(12), e2554.

And discussing their findings and any remaining questions about risk factors. 

Page 4 line 53-54 “Podoconiosis is more than Acquired Immunodeficiency disease, tuberculosis, malaria, and elephantiasis caused by Filarial worms” – more than what? Higher burden than these conditions?

Citation needed on page 5 lines 75-77 for “In areas where Podoconiosis is endemic, the majority of the community holds significant misconceptions about the cause, care, treatment, and prevention of the disease.”

Methods

on page 7 (k> 15 years) – is the “k” a typo?

Page 15 lines 133 “a blood specimen were collected and tested for Filiral worm to rule out lymphedema” - expect “lymphedema” should be “lymphatic filariasis”

Results- 

Tables 1 & 2 would be improved by confidence intervals for percentages, to assess whether these overlap between cases and controls. However, most of the information in these tables is duplicated in Table 3, which shows the variables which were significant on bivariate regression with crude odds ratios, and then in Table 4, which also includes adjusted ORs. I would question whether Tables 1-3 are all essential, or whether T4 alone would suffice, and non-significant variables (or all variables) could be shown in one supplementary table instead. The Results section is quite long (with some tables stretching over multiple pages) and shortening it would make the key results more prominent. 

on page 20, line 206- 207 “of these 30 (30%) of cases wore canvas whereas 129 (46.4%) of controls wore 207 leather shoes” – it’s not immediately clear why this comparison is drawn- suggest “among cases, the most common type of shoe was canvas, worn by 30.0%, while among controls the most common type was leather, worn by 46.4%)

In Table 2 check % for canvas shoes in controls- is 284 missing a decimal point?

Page 25 – where p values are very low, use <0.0001 rather than = 0.000 

Discussion- in addition to interpretation of results with reference to literature, it would be interesting to read about the public health implications of the work.

Reviewer #2: “Minor Revision”

**Summary and General Comments**

Reviewer #1: To my knowledge, this study is original in terms of its precise setting and research question. Although podoconiosis risk factors have been investigated at national level in Ethiopia*, the findings of the this study are somewhat different, potentially pointing to context-specific determinants of disease. 

The findings of the study are relevant to researchers, practitioners, and policy makers. In particular, findings on protective shoe wearing practices could inform health education and control.

Given the prevalence of podoconiosis in Ethiopia and its impact on health and economic participation, I anticipate this study to be of broad interest to people working in this setting and other podo-endemic areas. The paper is written in a way that is accessible to a broad base.

Given the previous larger-scale work of similar design in Ethiopia*, I would not describe this as outstanding, but I would say it is a strong paper with robust analysis, mostly clear interpretation, and interesting findings for the context. The manuscript is very well organised.

I have some concerns about the sampling procedures which I believe may have led to bias, but I think this could be accounted for with re-analysis, and that this study has potential for publication in PLoS NTDs.

The main conclusions are that age, gender, shoe wearing practices and family history are associated with the risk of podoconiosis. 

An association with age is expected (and has been previously demonstrated) for podoconiosis, and the authors describe their findings as consistent with those from the nationwide study- although the earlier study analysed age as a continuous variable and found a small but significant association whereas this one treats it as categorical and finds one age group (35-44) to be associated with a lower odds, so this is not directly comparable. 

The association between gender and podo was very strong (AOR of 3). The study says this was similar to that found in previous studies which are cited. However, reference 15 found an AOR for sex of 1.3; and ref 18 found an AOR of 1.4. 

Since the male:female ratio of cases in the study (1:1.55) is similar to the previous studies (1:1.38 in ref 17 and 1:1.43 in ref 18), I believe the biased sex ratio of controls led to an artificially inflated estimate of the association with gender. 

The nationwide mapping of podoconiosis in Ethiopia* did not identify any association with shoe wearing, so these findings are quite significant in terms of control initiatives for podo in this area.

It may be possible to correct for age- and sex-bias of controls by re-selecting a sub-set of controls, matched with cases on age-group and gender, and re-analysing the data. This would allow the authors to assess whether associations with the other variables remained, although it would leave a much smaller sample size. It would also make it impossible to get AORs for age and sex, but the age-sex distribution of cases could be compared to that from census data. This is quite a substantial re-analysis, but would correct the age-sex bias, which would be a significant improvement. However, if control sampling was not, then cases would still differ systematically from controls which would be a limitation to acknowledge. 

If the analysis is left as it is, I would suggest (although this may not be essential) including age as a continuous rather than categorical variable in the analysis. There seems to be no justification for treating it as categorical. This would not be difficult to implement but would require updating AORs. 

I think the paper would be strengthened by bringing in recommendations for public health practice into the discussion section, and by expanding the limitations section.

It would be useful for other researchers to be able to access the survey protocol.

There are no details on whether data are accessible. If they can be made public this would be of benefit to other researchers.

Reviewer #2: It is a well conducted and well reported study.

The content is too detailed for a journal. It has to be summarized to about 10-15 pages.

It was better if the authors mentioned the uniqueness of this study, comparing to other studies conducted in Ethiopia?

PLOS authors have the option to publish the peer review history of their article (what does this mean?). If published, this will include your full peer review and any attached files.

Reviewer #1: No

Reviewer #2: No
---

## [Decision Letter · Decision Letter 1]

25 Jun 2023

Dear Dr. Ali,

Thank you very much for resubmitting your manuscript "Determinants of Podoconiosis in Bensa District, Sidama Regional State, Ethiopia: A community based survey" for consideration at PLOS Neglected Tropical Diseases. As with all papers reviewed by the journal, your manuscript was reviewed by members of the editorial board and by several independent reviewers. The reviewers appreciated the attention to an important topic. Based on the reviews, we are likely to accept this manuscript for publication, providing that you modify the manuscript according to the review recommendations. Revisions to thoroughly address the concerns raised below are required.

Sincerely,

Gail Davey

Guest Editor

Victoria Brookes

Section Editor

Many thanks for your resubmission. The manuscript is much improved, but still requires a couple of further changes to fully address the issues raised by Reviewer 1.

Reviewer's Responses to Questions

**Key Review Criteria Required for Acceptance?**

**Methods**

-Are the objectives of the study clearly articulated with a clear testable hypothesis stated?

-Is the study design appropriate to address the stated objectives?

-Is the population clearly described and appropriate for the hypothesis being tested?

-Is the sample size sufficient to ensure adequate power to address the hypothesis being tested?

-Were correct statistical analysis used to support conclusions?

-Are there concerns about ethical or regulatory requirements being met?

Reviewer #1: This submission is a revised version of PNTD-D-22-01405R1. The authors have provided comprehensive responses to the comments raised.

Reviewer #2: The objectives of the study are clearly articulated with a clear testable hypothesis. The study design is appropriate to address these objectives. The population is clearly described and it is appropriate for the hypothesis being tested. The sample size is sufficient to ensure adequate power to address the hypothesis being tested. Correct statistical analysis has been used to support conclusions. The ethical issues are well addressed.

**Results**

-Does the analysis presented match the analysis plan?

-Are the results clearly and completely presented?

-Are the figures (Tables, Images) of sufficient quality for clarity?

Reviewer #1: Yes

Reviewer #2: The analysis presented matches the analysis plan.

The results are clearly and completely presented. The way the results organized are well detailed and well organized.

The statistical analysis results are well interpreted.

The figures and tables are good quality and clarity.

**Conclusions**

-Are the conclusions supported by the data presented?

-Are the limitations of analysis clearly described?

-Do the authors discuss how these data can be helpful to advance our understanding of the topic under study?

-Is public health relevance addressed?

Reviewer #1: The authors chose not to reanalyse their data to correct for the biased age distribution of controls, but have provided some explanation in the limitations section. I am still concerned that the sampling of controls has biased the estimate of the effect of age (the true odds ratio for podo in females versus males likely to be closer to 1.5 than 3). This is of concern because the presented AOR of 3 may be quoted in the future or included in systematic reviews/ meta-analyses without consideration of the impact of the sampling bias. With this in mind, I think the authors should be more transparent in the limitations section about how this bias is expected to have affected the results (beyond saying it “can bias the finding of this study”). For example- “we acknowledge that the preponderance of males in the control sample is likely to have led us to overestimate the association of podoconiosis with gender”. 

The limitations section also states that the authors used systematic random sampling. Systematic random sampling is intended to select sample data that represents a population. However, the systematic random sampling in this study was not applied to the study population, but to the kebele register, which was recognised to be biased as females were much less likely to be registered. I would therefore suggest removing “Even though we have employed systematic random sampling”, as some readers may interpret this as systematic random sampling of the population.

Reviewer #2: The conclusions are supported by the data presented.

The limitations of analysis are clearly described. The public health relevance including risk factors and potential prevention mechanisms has been addressed.

The authors discuss how these data can be helpful to advance our understanding of Podoconiosis.

**Editorial and Data Presentation Modifications?**

Reviewer #1: “The geographic information was downloaded to the computer program to generate a spatial 161 distribution of cases.” – this can be removed as the geographic mapping method is described fully later on. 

“Finding high odds of podoconiosis among participants with a history of podoconiosis” – a family history

“waking barefoot” (page 20 and 28) - walking

Reviewer #2: “Accept” with minor edits as shown bellow (It is also attached)

In Table 2- Type of shoes-canvas among control groups should be 28.4 % ( not 284). The same comment as the previously given one

Line 276 – The Adjusted OR should be 0.07 not 07 (the same comment was given earlier)

Line 339- I think the sentence ‘Finding high odds of podoconiosis among participants with a history of podoconiosis’ , should be stated as ‘Finding high odds of podoconiosis among participants with a family history of podoconiosis’

**Summary and General Comments**

Reviewer #1: (No Response)

Reviewer #2: It is a well conducted and well reported study.

Most of the comments have been well addressed.

Though the topic is not novel , there was no similar study in this study site. The determination of risk factors will help in the prevention and control of podoconiosis.

It would have been better if this case control study was properly matched.

PLOS authors have the option to publish the peer review history of their article (what does this mean?). If published, this will include your full peer review and any attached files.

Reviewer #1: No

Reviewer #2: No

Figure Files:

Data Requirements:

Reproducibility:

References

---

## [Editor Report · Decision Letter 2]

5 Jul 2023

Dear Dr. Ali,

We are pleased to inform you that your manuscript 'Determinants of Podoconiosis in Bensa District, Sidama Region, Ethiopia: A community based survey' has been provisionally accepted for publication in PLOS Neglected Tropical Diseases.

Best regards,

Gail Davey

Guest Editor

Victoria Brookes

Section Editor

<style type="text/css">p.p1 {margin: 0.0px 0.0px 0.0px 0.0px; line-height: 16.0px; font: 14.0px Arial; color: #323333; -webkit-text-stroke: #323333}span.s1 {font-kerning: none

</style>

---

## [Editor Report · Acceptance letter]

26 Jul 2023

Dear Dr. Ali,

We are delighted to inform you that your manuscript, "Determinants of Podoconiosis in Bensa District, Sidama Region, Ethiopia: A case control study," has been formally accepted for publication in PLOS Neglected Tropical Diseases.

Best regards,

Shaden Kamhawi

co-Editor-in-Chief

Paul Brindley

co-Editor-in-Chief
